# Iodine-Modified Ag NPs for Highly Sensitive SERS Detection of Deltamethrin Residues on Surfaces

**DOI:** 10.3390/molecules28041700

**Published:** 2023-02-10

**Authors:** Zhangmei Hu, Dandan Peng, Feiyue Xing, Xiang Wen, Kun Xie, Xuemei Xu, Hui Zhang, Feifei Wei, Xiaoke Zheng, Meikun Fan

**Affiliations:** 1The Analysis and Testing Center, Southwest Jiaotong University, Chengdu 610031, China; 2Faculty of Geosciences and Environmental Engineering, Southwest Jiaotong University, Chengdu 610031, China; 3Physical Science and Technology, Southwest Jiaotong University, Chengdu 610031, China; 4Sichuan Academy of Environmental Sciences, Chengdu 610041, China; 5Sichuan Zhongbiao Technology Co., Ltd., Chengdu 610052, China

**Keywords:** SERS, deltamethrin, pesticide residues, Ag NPs, iodide ion

## Abstract

It is essential to estimate the indoor pesticides/insecticides exposure risk since reports show that 80% of human exposure to pesticides occurs indoors. As one of the three major contamination sources, surface collected pesticides contributed significantly to this risk. Here, a highly sensitive liquid freestanding membrane (FSM) SERS method based on iodide modified silver nanoparticles (Ag NPs) was developed for quantitative detection of insecticide deltamethrin (DM) residues in solution phase samples and on surfaces with good accuracy and high sensitivity. The DM SERS spectrum from 500 to 2500 cm^−1^ resembled the normal Raman counterpart of solid DM. Similar bands at 563, 1000, 1165, 1207, 1735, and 2253 cm^−1^ were observed as in the literature. For the quantitative analysis, the strongest peak at 1000 cm^−1^ that was assigned to the stretching mode of the benzene ring and the deformation mode of C-C was selected. The peak intensity at 1000 cm^−1^ and the concentration of DM showed excellent linearity from 39 to 5000 ppb with a regression equation I = 649.428 + 1.327 C (correlation coefficient R^2^ = 0.991). The limit of detection (LOD) of the DM was found to be as low as 11 ppb. Statistical comparison between the proposed and the HPLC methods for the analysis of insecticide deltamethrin (DM) residues in solution phase samples showed no significant difference. DM residue analysis on the surface was mimicked by dropping DM pesticide on the glass surface. It is found that DM exhibited high residue levels up to one week after exposure. This proposed SERS method could find application in the household pesticide residues analysis.

## 1. Introduction

To kill or control pests such as rodents, insects, fungi, bacterium and other organisms, pesticides have been widely used in almost all households. The indoor pesticide residues will result in exposing to the polluted environment and have harmful effect on human health [1,2,3]. According to EPA, there are three major sources in households, namely contaminated soil/dust coming from outside the home, pesticide containers that stored home, and household surfaces [4]. Thus, it is essential to develop methods that can sensitively monitor surface bound pesticides to minimize the possible exposure risk to humans. DM is a synthetic organic pyrethroid insecticide commonly used as an insecticide in homes to control mosquitoes and cockroaches after spraying [5,6,7]. Reports show that pyrethroids can be absorbed by the body through inhalation and skin contact with contaminated surfaces resulting in toxic damage to the human nervous, immune, metabolic system, inducing cancer and deformity, etc. [5,8,9,10]. Consequently, it is necessary to establish quantitative method for insecticide deltamethrin (DM) residues in the household environment.

So far, various analytical techniques have been employed to detect the deltamethrin (DM) residues, including chromatography (GC and HPLC), chromatography-mass spectrometry (GC-MS and HPLC-MS), fluorescence, chemiluminescence and colorimetric methods [11,12,13,14,15,16,17]. Although those methods have good sensitivity, many of them are difficult to be deployed in field applications due to bulky instrumentation.

Surface-enhanced Raman scattering (SERS) has been widely known for pesticide analysis with high specificity (capable of providing fingerprint-alike information) [18,19], high sensitivity (possible of single molecules analysis) [20], fast detection [21] and portability [22,23]. One of the keys to achieving high sensitivity in SERS analysis is to effectively bring the analyte close enough to the SERS substrate. Various methods have been reported to be able to maximize the SERS signal through this approach. Among them, certain inorganic salts showed considerable signal enhancing efficiency. Halide ions were considered effective co-adsorbates to enhance the SERS sensitivity, and many groups, including us [24,25], had been working in this field intensively. For example, polycyclic aromatic hydrocarbons [26], antibiotics [27], drugs [28] have been shown to enhanced SERS signals in the presence of certain halogen ions. Simultaneously, the mechanism of co-adsorbates enhanced SERS sensitivity was also investigated. It was widely believed that the synergistic effects among the target molecules surface, and certain halide ions could realize high SERS sensitivity. However, the halide ions may trigger unstable aggregation states resulting in poor reproducibility, which will affect its reliability as a routine quantitative method and even prevent its widespread use. In our previous work [29], signal repeatability in SERS has also been greatly improved by simply forming the liquid free-standing film of SERS active Ag NPs. The relative standard deviation of SERS signal on an FSM was found to be less than 10%. Most importantly, the preparation of FSM substrate is simple and straightforward, and can be easily deployed onsite [23].

Here, a highly sensitive FSM SERS method based on iodide modified silver nanoparticles (Ag NPs) was proposed for quantitative detection of insecticide deltamethrin (DM) residues in the solution phase and on surface. The SERS method took advantage of the reproducible FSM [29], and the excellent sensitivity for DM with iodide modified Ag NPs. DM solution was drop-casted onto the glass surface and left in air for different times. SERS analysis was performed by rinsing the contaminated glass slides with methanol. The result showed that the pesticide residue was relatively high for at least one week.

## 2. Results and Discussion

### 2.1. SERS Substrate

The characterization of the Ag NPs was shown in Figure 1a. It is clear that the average diameter of the Ag NPs was about 50–70 nm. It is also clear in Figure 1b that the LSPR absorption of Ag NPs and Ag-I NPs appeared at 404 nm and 415 nm, respectively. A red shift of the LSPR band of the Ag NPs but almost no broadening was observed. The former was believed to be the result of surface modification of Ag NPs, the latter demonstrated that the Ag-I NPs has good stability (not aggregating) in the suspension [30]. In other words, the iodide formed a layer on the Ag NPs surface, as suggested in the literature.

### 2.2. SERS Sensitivity Optimization

Figure 2a illustrates how iodide can be used to improve the SERS signal of DM using the FSM SERS method. It is clear that with bare Ag NPs, 1 ppm of DM showed almost no SERS signal, while in the presence of I^−^, a clearly visible SERS signal at 563, 1000, 1165, 1207, 1735, and 2254 cm^−1^ was shown. This strongly suggested the effectiveness of I^−^ in SERS signal boosting. The assignments of the bands for the molecular structure of DM are listed in Appendix A. Compared with the characteristic peaks of 563 or 2253 cm^−1^, the 1000 cm^−1^ was the highest peak for the DM, as shown in Figure 2a line 1. Therefore, 1000 cm^−1^ was selected for further analysis. In Figure 2b, the spectra of Ag-I NPs with and without methanol together with the normal Raman of DM were shown. The spectra indicate Ag-I NPs with/without CH_3_OH had no characteristic peaks in the above noted positions, demonstrating that the observed bands are all from DM. This was further proved from Figure 2b line 3, the normal Raman of DM, where similar bands were shown.

We then explored the SERS improving performance of different kinds of halide ions Cl^−^, Br^−^ and I^−^ using 1 ppm DM as the model analyte. As shown in Figure 3a, no obvious SERS signal DM was observed for Cl^−^ modified Ag NPs. In terms of Br^−^, barely visible SERS signal was found at 563, 1000, and 2254 cm^−1^, respectively. The results in Figure 3a indicated that only I^−^ showed observable SERS enhancement. Thus, we decided to use iodine ion for DM detection in this work. It was suggested that halide ions can enhance the SERS signal by forming a coating layer over the Ag NPs upon ligand exchanges. As shown in Appendix A, the largest red shift of UV spectra was caused by iodide ion, which further confirmed that the optimum halide ion species is generally target specific [24,27].

The sequence of preparation steps as well as the concentrations of KI were investigated. The corresponding result was shown in Appendix A. The best SERS signal was produced by first combining the KI with the Ag NPs and then adding the DM. This correlates well with the literature [25], since it is widely believed that the behind mechanism of halide ions improved SERS was due to the replacement of the adsorbed citrate ions by halide ions (I^−^ in this case), which then makes position for analyte molecule adsorption. By varying the concentrations of KI from 0.025 M to 0.5 M (Figure 3b), it turned out that the SERS signal gradually reduced at concentrations lower or higher than 0.1 M. The 0.1 M of KI was the best for DM detection in this method. It is believed that an excessive amount of KI may cause the colloidal solution unstable and eventually lead to the precipitation of NPs, thereby reducing SERS signal intensity [31]. Thus, 5 μL of 0.1 M KI were mixed with 20 μL concentrated Ag NPs, followed by 10 μL DM, which is used in this work.

### 2.3. Analytical Merits of the SERS Method

To examine the reproducibility of the FSM SERS platform, SERS spectra of DM obtained randomly from 10 sites of the FSM were shown in Figure 4a, which could reflect that SERS signal of DM had good reproducibility. Meanwhile, eight FSM SERS substrates were prepared as shown in Figure 4b, where the peak intensities of DM were calculated at 1000 cm^−1^. It can be observed that the RSD of each FSM was within 12.7% (n = 10) in Appendix A. Here, we used the FSM approach to realize the reproducible SERS analysis.

The SERS spectra range from 500 cm^−1^ to 1400 cm^−1^ of DM at different concentrations from 0 to 5 ppm were recorded. The insert picture in Figure 5 illustrates that how the SERS signal of DM at 563 cm^−1^ and 1000 cm^−1^ increased with increasing the concentration of associated targets. The peak intensity at 563 cm^−1^, however, was drowned by background noise when the concentration fell to 156 ppb and no longer changed with the concentration. However, for the peak intensity 1000 cm^−1^, When the concentration is as low as 39 ppb, the SERS signal is still detectable. In further work, the 1000 cm^−1^ as the maximum enhancement peak was selected as the characteristic peak for the quantitative analysis of DM.

Figure 5 illustrated that the linear regression equations established for the peak intensity of 1000 cm^−1^ and the DM concentration, and the results showed that the peak intensity had a good linear relationship (I = 649.428 + 1.327 C) with the DM concentration in the range of 0 to 5 ppm and the R^2^ was 0.991. According to the literature [32,33], the limit of detection (LOD) was 11 ppb calculated by 3σ/slope, where σ is the standard deviation of the blank samples.

### 2.4. Verification of the FSM SERS Method for DM Pesticide Detection

The developed SERS method was applied for determination of the commercial DM product from Bayer. As shown in Appendix A, the characteristic peaks of DM at 563 and 1000 cm^−1^ can be identified in commercial product. DM residues samples in solution phase were analyzed by the proposed SERS and HPLC method. The calibration curve for DM using HPLC is shown in Appendix A. Experiment results showed that there is no significant difference between the two methods (F < F crit in Appendix A). These results demonstrated that the quantitative analysis using the proposed FSM SERS method was reliable.

### 2.5. Comparison of the SERS Method with Other Reported Methods

The comparison of different methods for DM detection was introduced in Table 1. For the GC, fluorescence, chemiluminescence, and colorimetric methods, the LODs were all above 100 ppb. However, for this FSM SERS method, the calculated LOD was 11 ppb, which means our method has better sensitivity. In a word, this method can achieve the trace detection of DM by regulating SERS Ag substrate with iodide ions.

### 2.6. Monitoring DM Insecticide Residues on the Glass Surface

The change in pesticide residue level of DM after drop-casting on the glass surface within one week was shown in Figure 6. Obviously, the effective amount of the drug did not decrease significantly. Even after a week, the pesticide residue still was 93.5%. Overall, the pesticide residue was relatively high. According to previous reports [36], it was mostly due to DM’s non-volatile nature and chemical stability. In addition, the surfactant and other components also affect the active stability and physicochemical characteristics of the emulsion pesticide.

## 3. Materials and Methods

### 3.1. Reagents and Materials

All chemicals used in this work were at least analytical grade. Deltamethrin (C_22_H_19_Br_2_NO_3_, 98.6% purity), sodium citrate (99%), silver nitrate (99.999%), potassium iodide (KI), potassium chloride (KCI), and potassium bromide (KBr) were purchased from Sigma-Aldrich (Shanghai, China). HPLC grade methanol was purchased from Honeywell (Morris Plains, NJ, USA). The DM insecticide in suspended dosage was from a supermarket, and the effective concentration of deltamethrin in the product was 2.5 × 10^4^ mg/L. The customized stainless steel perforated template has a hole with a diameter of 5 mm, and the thickness of the perforated template is 0.6 mm [29]. De-ionized water (18 MΩ-cm) used all through the experiments was obtained from a water purification system (Pure Technology Co., Ltd., Chengdu, China).

### 3.2. Instrumentation

The UV-Vis spectra were recorded on a portable spectrophotometer (MAYA2000Pro, Ocean Optics, Orlando, FL USA). The SEM images were obtained with the 7800F Field emission scanning electron microscope (FE-SEM, JEOL, Tokyo, Japan). The SERS spectra were obtained by an HR-Evolution Raman microscope (HORIBA, Kyoto, Japan). The 532 nm laser, 50× objective (N.A.= 0.5), 600 gr/mm holographic grating, and 100 μm pinhole were used throughout the experiment.

### 3.3. Preparation of Ag NPs

Ag NPs were prepared according to previous report [29,37]. In brief, 0.017 g of AgNO_3_ was dissolved in 100 mL ultrapure water and heated to boil. Later, 2 mL 1% sodium citrate was quickly added. The stirring and heating were kept for an hour. After that, heating was discontinued, stirring was kept until it cooled to room temperature. Finally, the concentrated Ag NPs were prepared by centrifugation in a ratio of 1 mL to 10 μL at 10,000× *g* for 10 min.

### 3.4. Preparation of FSM SERS Substrate

Firstly, 20 μL concentrated Ag NPs was thoroughly mixed with 5 μL of KI by vortexing for 5 s in a 1.5 mL centrifuge tube. Next, 10 μL DM was added to the tube followed by vortexing for another 5 s. Then, a 15 μL mixed suspension was placed on Teflon tape to form a spherical droplet. A stainless-steel mode was used to pick up the drop and form the FSM for SERS analysis (Figure 1).

### 3.5. The DM Solution Phase Sample Treatment

A 1000 ppm stock solution was prepared by dissolving DM in methanol. It was then diluted to the desired concentration with methanol.

The DM solution phase samples were prepared by dluting the commercial DM products to the desired concentration with methanol. Filtrating the sample with 0.45 um microporous membrane was carried out before SERS analysis.

### 3.6. HPLC Analysis

The mobile phase was methanol with a 1.0 mL/min flow rate, and the injection volume was 5 μL. A UV detector at 238 nm was used for the analysis. The column oven (Thermo Fisher Scientific, Shanghai, China) was set at 30 °C. The injection volume was 20 μL. The HPLC calibration curve for DM can be seen in Appendix A. HPLC was used to analyze these simulated samples.

### 3.7. Surface Residue Analysis

The 20 mL original DM pesticide suspension with a high concentration (2.5 × 10^4^ ppm) was diluted with water to 1300 mL to prepare the pesticide spray solution according to the direction of the commercial product. Then, 10 μL pesticide spray solution was drop-casted onto the glass surface for monitoring pesticide residues at different times. The detailed process can be found in the supporting information (Appendix A). The contaminated glass slides were left in the ambient environment to mimic the indoor environment. The glass slides were collected at 12 h intervals after dropping for one week and subjected to SERS analysis.

To collect the DM residue, the contaminated glass slides were rinsed with 3 mL methanol, which was then placed in a water bath and purged with nitrogen for concentrating to 1 mL. Finally, it was filtrated with 0.45 μm microporous membrane before SERS analysis. Data analysis was performed using Origin Pro 2023 software. The following mathematical equation is used for computing the residual percentage.
Residual percentage = Ci/C1 × 100%
where Ci represents DM residual concentration of each time interval, C1 represents the concentration of pesticide residue initially.

## 4. Conclusions

In this work, a fast, sensitive, and reliable SERS method for the detection of DM was developed with iodide-modified Ag NPs as SERS substrate. The DM within 39 ppb to 5 ppm range increased with increasing SERS signal in a linear manner, with the calculated LOD of 11 ppb. DM emulsion from Bayer was successfully analyzed by this method compared with HPLC, where the results showed no significant difference. Moreover, DM emulsion was dropped onto the glass surface, the insecticide’s residues on the glass surface were monitored for one week, and it was found to demonstrate minimum loss during the time in ambient air. The result may imply long time exposure of the DM pesticide in the household scenario.

## Data Availability

Not applicable.

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
