# Peer review of "Iodine-Modified Ag NPs for Highly Sensitive SERS Detection of Deltamethrin Residues on Surfaces"

_molecules, 2023, doi:10.3390/molecules28041700_

Round 1
Reviewer 1 Report
This work is of great practical significance.
1. Figure 1 is not clearly marked, and the experimental data of Ag-I NPs with DM should appear in figure 1. It can not be concluded from Figure 1 that Ag-I NPs is better than Ag NPs as the SERS substrate.
2. Lack of characterization experimental data of Ag NPs and Ag-I NPs, and lack of stability experimental data of Ag-I NPs.
3. There is no Raman spectrum analysis of DM. How to select the characteristic peak for qualitative and quantitative research is important in the work of analysis. In this work, why is 1000 cm-1 selected as the characteristic peak?
4. How is LOD calculated? No data to support the calculation process? Lack of the experimental data of standard addition recovery rate experiment?
5. Figure 6 is not found in the submission?
Author Response
Dear Reviewer,
We greatly appreciate your time and consideration concerning our manuscript entitled “Iodine modified Ag NPs for highly sensitive SERS detection of deltamethrin residues on surfaces” (Manuscript ID molecules-2129941). After carefully reading the comments, we have made changes to the manuscript.
This manuscript is a revised edition of the original submission “Iodine modified Ag NPs for highly sensitive SERS detection of deltamethrin residues on surfaces”. The original version has been reviewed by four reviewers. The concerns of four reviewers have been addressed in this manuscript.
We trust that this revised version has addressed your concerns.
Thanks for you Comments and Suggestions. Attached is a letter to answer your relevant questions. Hope the reply can addressed your concerns.
Sincerely yours,
Meikun Fan
Faculty of Geosciences and Environmental Engineering – Southwest Jiaotong University
Email: mkfan@swjtu.edu.cn

Reviewer 2 Report
Hu et al., proposed a SERS-based analytical method for quantitative analysis of insecticide deltamethrin (DM). The manuscript can be accepted after a minor revision.
1) Short forms can be avoided. For instance, “it’s found” can be written as “it is found.” “It’s” can be written as “it is.”
2) The authors can rectify spelling and punctuation mistakes. For example, Line number 37, “Consequently, It is” can be corrected as “Consequently, it is.” Line 43, “filed” should be corrected as “field.”
3) Schematic diagram 1, “Toflon tape” should be corrected as “Teflon tape.”
4) Photographs of SERS substrates can be included in the manuscript.
5) In Figure 4, “5000 ppb, 2500 ppb, 1250 174 ppb, 625 ppb, 313 ppb, 156 ppb, 78 ppb, 39 ppb” can be written as “5000, 2500, 1250, 174, 625, 313, 156, 78, and 39 ppb.”
Author Response
Dear Reviewer,
We greatly appreciate your time and consideration concerning our manuscript entitled “Iodine modified Ag NPs for highly sensitive SERS detection of deltamethrin residues on surfaces” (Manuscript ID molecules-2129941). After carefully reading the comments, we have made changes to the manuscript.
This manuscript is a revised edition of the original submission “Iodine modified Ag NPs for highly sensitive SERS detection of deltamethrin residues on surfaces”. The original version has been reviewed by four reviewers. The concerns of four reviewers have been addressed in this manuscript.
We trust that this revised version has addressed your concerns.
Thanks for you Comments and Suggestions .Attached is a letter answering your relevant questions. Hope the reply can addressed your concerns.
Sincerely yours,
Meikun Fan
Faculty of Geosciences and Environmental Engineering – Southwest Jiaotong University
Email: mkfan@swjtu.edu.cn

Reviewer 3 Report
In my opinion, this manuscript is not well organized. In addition, English writing also needs to be improved. The image is not clear. The error bar reflects that the data may be abnormal. The literature was cited irregularly. The supporting data should be placed in the manuscript, because the amount of data in the manuscript is too small. In summary, this manuscript is a failure. It cannot be considered in its present form.
Author Response

(The authors gave the same response as above.)

Reviewer 4 Report
Reviewer #1: Comments on molecules-2129941
Title: Iodine modified Ag NPs for highly sensitive SERS detection of deltamethrin residues on surfaces
By: Meikun Fan
_______________________________________
The paper presents a highly sensitive liquid freestanding membrane (FSM) SERS method based on iodide modified silver nanoparticles (Ag NPs) was developed for quantitative detection of insecticide deltamethrin (DM) residues in solution phase samples and on surfaces with good accuracy and high sensitivity. It is a topic of interest to the researchers in the related areas,and the paper is well organized, and the improved minor version before accepted publication in Molecules. My detailed comments are as follows:
Q1. What is the selection criteria for halide ions and Ag NPs? Are the other NPs or target also effective in the proposed system?
Q2. FSM SERS substrate modified with KI showed the best SERS response. Why? Give further discussion and analysis. It is proposed to briefly explain in the article the principle that halogen ions can enhance the SERS
Q3. There are some printing and grammar error, and please check the manuscript throughout carefully. Such as,
Line 118, " after droppig ";
Fig. 2 (b) Horizontal and vertical coordinates suggested to be bolded;
Figure 4 can't be seen very well.
Q4. Please describe the performance of the developed sensor system in this study by comparing with other DM sensors based on SERS technique. And adding the more references including 2022 is recommended.
Q5. Line108 - 110, it was mentioned that " the commercial DM product contains surfactant was diluted with methanol and subject to filtration with 0.45 μm microporous membrane before SERS and HPLC analysis.” How about the matrix effect for such a commercial DM product?
Author Response

(The authors gave the same response as above.)

Round 2
Reviewer 3 Report
I have no any questions.
Author Response
Thank you